# Psychiatric Clinical Placement: Agony for College Nursing Students in South Africa

**DOI:** 10.3390/ijerph20043545

**Published:** 2023-02-17

**Authors:** Thingahangwi Cecilia Masutha, Mary Maluleke, Ndidzulafhi Selina Raliphaswa, Takalani Eldah Thabathi, Mphedziseni Esther Rangwaneni, Ndivhaleni Robert Lavhelani, Duppy Manyuma, Lufuno Martha Kharivhe, Tshinanne Gladys Thandavhathu

**Affiliations:** Department of Advanced Nursing, University of Venda, Private Bag X5050, Thohoyandou 0950, South Africa

**Keywords:** clinical placement, agony, college, psychiatric hospital, student nurse

## Abstract

(1) Background: Clinical placements in psychiatric nursing science (PNS) are as important as other placements in any discipline in nursing education as they allow students to correlate theory to practice. Nursing students’ absenteeism has become a grave concern in psychiatric institutions in South Africa. This study investigated the clinical factors leading to student nurses’ absenteeism in the Limpopo College of Nursing during the psychiatric nursing science clinical placement. (2) Methods: A quantitative approach using a descriptive design was followed, where 206 students were purposively sampled. The study was conducted at the Limpopo College of Nursing situated in Limpopo Province, with five campuses offering a four-year nursing program. College campuses were used to access students since this was an easy way of accessing them. Data were collected through questionnaires of structured questions and analyzed using SPSS version 24. Ethical considerations were adhered to throughout. (3) Results: The correlation between clinical factors and absenteeism was determined. The main reported factors for absenteeism were student nurses being treated as a workforce in the clinical areas; shortage of staff in the clinical areas; inadequate supervision of student nurses by professional nurses; and being inconsiderate of student nurses requests for days off in the clinical area. (4) Conclusions: The findings revealed that student nurses’ absenteeism is caused by different factors. The Department of Health should find a way of not overworking students due to a shortage of staff in the wards but consider them for experiential learning. A further qualitative study should be conducted to develop strategies to mitigate student nurses’ absenteeism in psychiatric clinical placement.

## 1. Introduction

Absenteeism refers to being away from scheduled classes and training experiences irrespective of the reasons [1]. The Irish Nursing Council [2] stipulates that nursing students must obtain 100% of clinical placements each year, and those who still need to meet this requirement may not progress to the next level of study. On the other hand, in Northern Europe, Finland tolerates 95% of attendance to meet the requirements of the course, allowing only 5% of absenteeism by nursing students. Similarly, the Nigerian University of Abadan requires nursing students to acquire 75% attendance before they can sit for the examination of a specific subject [3]. According to the South African Nursing Council [4], if a student does not obtain 80% (320 h in level III and 400 h in level IV) of hours in psychiatric nursing science for clinical exposure, that student is not allowed to do the final clinical examination of a subject.

The Limpopo College of Nursing provides training to student nurses to become professional nurses. Before the commencement of training, nursing students in Limpopo Province enter a fixed-term appointment contract as trainee nurses. According to the Limpopo College of Nursing contract [5], there are a certain number of hours in theory and clinical areas that student nurses are expected to accumulate in levels III and IV of psychiatric nursing science. This contract aligns with the South African Nursing Council [4] requirements that student nurses are expected to obtain 80% (320 h in level III and 400 h in level IV) of the total hours of 720 in psychiatric nursing science for clinical exposure. Failure of this is regarded as absenteeism.

For the past few years, during the clinical accompaniment of students, the principal author has observed a high absenteeism rate among level III and level IV student nurses in the Limpopo College of Nursing during psychiatric nursing science clinical placement. Magobolo and Dube [6] found that 97.3% of nursing students were absent from clinical areas, and [3] indicated that in Nigeria, absenteeism among nursing students from clinical areas and lectures is a significant problem as most students do absent themselves most of the time. Similarly, ref. [7] also indicated that absenteeism among university and college nursing students is a challenge. In Ethiopia, a study conducted at universities by the authors of [8] indicates that many higher nursing institutions have strict policies regarding mandatory attendance during lectures, laboratory, and practical sessions by nursing students. Despite these policies, among nursing students, absenteeism is an ongoing problem in these universities.

Several factors might contribute to student nurses’ absenteeism while allocated to psychiatric clinical areas. In Iran, ref. [9] developed strategies to reduce absenteeism from students’ viewpoint in mental institutions, revealing negative views of people with mental illnesses as being dangerous and harmful. This was another issue students brought up during the interviews. Some students reported that they were terrified of being physically abused by patients. Van Rensburg [10] indicates that student nurses are primarily absent from clinical settings because they fear contracting communicable diseases found in the wards or if they are already infected, particularly in clinical areas. Many causes of absenteeism are due to personal illness or family issues, for example, but also can often be due to other factors, such as poor working environment or workers who are not devoted to their work [11]. Furthermore, ref. [12] reports that most students agreed that problems experienced in the clinical area, such as staff shortage, being treated as a workforce, non-supervision by qualified staff members, and large patient numbers, are why they are absent.

Absenteeism significantly impacts student nurses because, according to the South African Nursing Council [4], a student who has obtained less than 80% of clinical hours in a subject is not allowed to sit in for the final examinations of a subject. The same applies to theoretical examinations. As a result, the student may have to repeat a level for a year, leading to fewer students graduating at the end of four years, hence the shortage of professional nurses in the country. Absenteeism can lead to an extension of training and termination of student nurses from the program [2].

Nawaz et al. [13] indicated that there is no gaining of knowledge and skills for achieving personal and professional goals if students are absent from the classroom and clinical areas. Nawaz et al. [13] further indicated that absenteeism has a negative impact on students’ performance and prolongs the length of their study. It also portrays the low level of interest and motivation for studies. Similarly, ref. [14] shows that absenteeism has an adverse effect on the quality of nursing care that students must render. Randa [1] said that absenteeism causes the poor application of skills by the students, and it inhibits students from acquiring appropriate information and contact with relevant materials required for effective learning, and also is associated with poor academic performance, unprofessional conduct, and inadequate socialization within the profession.

A study was conducted at Mania University of Egypt by the authors of [15] about the influencing factors of absenteeism among nursing students. Another study was conducted in the college of nursing of India by the authors of [14] about nursing students’ absenteeism in class/clinics; on the other hand, a study about the determining factors influencing absenteeism among nursing students at the University of Lahore in Pakistan was conducted by the authors of [13]. Another study was conducted by the authors of [1] at the Universities of Usmanu DanFodiyo and Ahmadu Bello of Nigeria about the personal factors that influence absenteeism among student nurses in classes. Baloyi [16] conducted a study about nursing students’ absenteeism in the theory classroom of the Limpopo College of Nursing in South Africa. All of the above studies were conducted to investigate the absenteeism of nursing students from other countries, universities, and other nursing colleges, mostly in theory and not in psychiatric clinical areas. Therefore, this study investigated the psychiatric clinical factors contributing to student nurses’ absenteeism during a psychiatric clinical placement at Limpopo College of Nursing in South Africa.

## 2. Materials and Methods

### 2.1. Study Design

A quantitative approach using a descriptive design was followed [17]. A descriptive research design was used in this study via self-report questionnaires to describe clinical factors leading to absenteeism of level III and level IV student nurses during psychiatric nursing science clinical placement at the Limpopo College of Nursing.

### 2.2. Study Setting, Population, and Sampling Strategy

The study was conducted in three districts of Limpopo Province, namely the Mopani, Vhembe, and Capricorn districts, as the three specialized psychiatric hospitals are situated within these districts. In addition, the study included three specialized psychiatric hospitals in Limpopo Province, namely Evuxakeni, Hayani, and Thabamoopo hospitals, where three campuses (Giyani, Thohoyandou, and Sovenga) of the Limpopo College of Nursing place students of level III and level IV. The Evuxakeni Psychiatric Hospital is situated on the Giyani main road, approximately 5.3 km outside of the town of the Greater Giyani sub-district, which is made up of rural areas with a population of different cultures. Sotho-, Tsonga-, and Venda-speaking people are the dominant groups. It is surrounded by a shopping mall, district offices, Giyani local municipalities, and schools. The Hayani Psychiatric Hospital is situated in the Sibasa rural area on the main road, next to the city of Thohoyandou. The Thulamela Local Municipality, Vhembe district offices, and different malls surround it. Venda-speaking people dominate it. The Thabamoopo Psychiatric Hospital is situated in the Lepelle Nkumpi Local Municipality in the rural area of Lebowakgomo, southeast, approximately 50 km from Polokwane. The inhabitants speak Sepedi as their first home language. The small villages of Lebowakgomo surround it. Student nurses travel from the Giyani Campus by bus to the Evuxakeni Hospital, which is 5.3 km away. The distance from Thohoyandou Campus to the Hayani Hospital is 13.6 km, and it is 49.5 km from the Sovenga Campus to the Thabamoopo Hospital. The University of Limpopo and the University of Venda also use the Thabamoopo Psychiatric Hospital and the Hayani Psychiatric Hospital to place student nurses for their practical studies. The setting for data collection was a classroom for each level on each campus.

The respondents were visited in their respective classrooms to recruit them one day before the study. The potential respondents were informed about the purpose and nature of the study as outlined in the respondents’ information sheets. The respondents were also informed that they could withdraw from the study at anytime. Informed consent was obtained from the respondents before participating in the study.

Sampling occurred in three stages, namely, sampling of districts, sampling of campuses, and sampling of respondents. Three of the five districts in Limpopo Province were purposively selected (Vhembe, Mopani, and Capricorn) for the study, as the three specialized psychiatric hospitals and campuses of the Limpopo College of Nursing are situated in these districts.

Campuses and respondents were sampled purposively based on the study level and the following criteria: only campuses that place students in specialized psychiatric hospitals for clinical practice and train level III and IV students. A total population of 206 respondents, who had previously absented themselves from the psychiatric clinical area, was recruited and sampled purposively from the three campuses. In level III, there were 81 students. In level IV, there were 125 students.

Permission was sought from the following stakeholders: the University Higher Degree Committee, the Research Ethics Committee of the University of Venda (SHS/18/PH/14/1406), the Limpopo Province Department of Health, and the Vice Principals of Limpopo College of Nursing campuses. The environment in which questionnaires were completed was safe for respondents since it was in their usual classroom. Therefore, no harm came to the respondents. The respondents were reassured of confidentiality. The researcher respected any agreements made with them, including punctuality per time agreed upon. The respondents were requested not to write their names on the questionnaires.

### 2.3. Data Collection

The instrument was pre-tested for 10% (23) of students who did not participate in the main study but shared the same characteristics as the main study population. There were nine students from level III and fourteen from level IV, and consent was obtained. The completed questionnaires were presented to the supervisor and colleagues in a workshop for validity, and gaps were identified and rectified. According to the findings, the pre-test questionnaire had Section 1 of the demographic data, which included marital status, and it was debated. The results revealed no connection between students’ absenteeism and marital status. The section was removed from the questionnaire. The pre-test questionnaire had four aspects to indicate students’ answers: agree, strongly agree, neutral, disagree, and strongly disagree. Changes were made, and the corrected questionnaire only had two aspects (yes and no). Some of the questions were two-in-one, which were also corrected. Some questions were ambiguous in the pre-test questionnaire, and straightforward questions replaced them.

Data collection was conducted using a questionnaire with different views from the principal author and the supervisors. The same questionnaires were used with all groups involved in the study and were used the same way for all respondents to ensure reliability [18].

Questionnaires consisted of 2 questions on demographic information: the sample characteristics and the number of days respondents were absent from PNS clinical areas. They also consisted of 4 questions focused on factors in the clinical area that lead to student nurses’ absenteeism: Are students treated as a workforce? (Yes and no columns were provided as the answers to choose from). Is there a staff shortage in the psychiatric clinical area? Is there adequate supervision in psychiatric wards by professional nurses? Are students’ requests for off-duties considered or not?

The questionnaires were administered in the respondents’ classrooms. The respondents filled in the questionnaire while the researcher was waiting in the classroom to provide support and clarify where respondents did not understand. The respondents were given enough time (30 min) to complete the questionnaires since this method quickly obtained data from a large group. The collected data were presented to several panels of research seminars to check the content validity. The researcher was objective and did not allow subjective feelings and thoughts to influence data collection questions.

### 2.4. Data Analysis

Data were analyzed using Statistical Package for Social Sciences (SPSS) software version 24. Descriptive and inferential statistics were performed to analyze data.

Responses from respondents were compiled into percentages and presented in charts and tables. This was performed to facilitate easy analysis and understanding of data of the study that sought to investigate the clinical factors leading to student nurses’ absenteeism in the Limpopo College of Nursing during psychiatric nursing science placements.

## 3. Results

About 206 questionnaires were distributed to respondents who consented to participate in the study, and 204 questionnaires were satisfactorily filled in and returned. Thus, the response rate was 99%.

### 3.1. Demographic Information

#### Sample Characteristics

Section 1 sought to gather the demographic information of study respondents, and questions on issues such as age, gender, and levels of study were asked.

Of 204 respondents, with regard to age, 186 (91.2%) were of the age group 19–25 years, while 16 (7.8%) were in the age group 26–32 years. However, only two (1%) respondents were above 32 years old. It is evident from the findings that most of the nursing students were 19–25 years old, while a few students were above 32 years old. Of the 204 respondents, the majority, 156 (76.5%), were female students, while 48 (23.5) were male students, as indicated. From these findings, most nursing students who participated in this study were females.

Of the 204 respondents, the majority, 120 (58.8%), were in level III, while 84 (41.2%) were in level IV of their nursing studies. Most of the nursing students who were respondents to this study were in level IV.

According to the chi-square test, regarding the age of the respondents, according to the linear-by-linear association, there was a *p* value of 1.046 and df of 1 with asymptotic significance (two-sided) of 0.306. The minimum expected count was 10 and had an expected count of less than 5. This shows that most students who were absent were between the ages of 19–25. Regarding gender, according to Fisher’s exact test, the exact significance (two-sided) was 0.194, the exact significance (one-sided) was 0.122, and they had an expected count of less than 5. The minimum expected count was 4.26; therefore, more females were absent than male students. There was an association between the age of respondents and the absenteeism of student nurses, as well as the gender of the respondents and the respondents’ absenteeism.

### 3.2. Psychiatric Clinical Factors

Section 2 of the questionnaire sought to collect information on the views of respondents regarding student nurses’ absenteeism in a psychiatric clinical placement in South Africa. The following factors were identified: students are treated as a workforce in the clinical area, a shortage of staff in the ward, inadequate supervision of students by professional nurses, and students’ requests for days off are not considered.

Of the 204 respondents who responded to this question, 105 (51.4%) respondents indicated that students are treated as a workforce. In contrast, 99 (48.6%) indicated that students are not treated as a workforce in the clinical areas. Based on these findings, it can be said that in some clinical regions, students are treated as a workforce, while other clinical areas do not treat students as a workforce. According to the information respondents provided, seen in Figure 1, some respondents were absent due to being treated as a workforce in the clinical area.

Of the 193 respondents who responded to this question, 129 (66.8%) of the respondents indicated a shortage of staff in the wards, while 64 (32.2%) indicated no staff shortage. It is evident from the findings that there was a shortage of staff in the wards, which may have caused students to be absent from the PNS clinical areas because they may have been left alone with a huge patient load. Based on the information of respondents provided in Figure 2, most respondents were absent due to a shortage of staff.

Of the 196 respondents who responded to this question, 118 (60.2%) respondents reported no adequate supervision, while 78 (39.8%) indicated adequate supervision by professional nurses. From the findings, it is evident that supervision for students by professional nurses in the clinical areas is rarely provided, as most respondents indicated, as presented in Figure 3. Based on the information of respondents provided in Figure 3, most respondents were absent due to inadequate supervision by professional nurses.

Of the 197 respondents who responded to this question, the majority, 112 (56.9%) respondents, indicated that their request for days off was not considered, while 85 (43.1%) indicated that their request for days off was considered. Based on the information of respondents provided in Figure 4, most respondents were absent due to inconsideration of a request for days off.

## 4. Discussion

The aim of the study was to investigate the clinical factors leading to student nurses’ absenteeism in the Limpopo College of Nursing during the psychiatric nursing science clinical placement. The findings revealed the following as being psychiatric clinical factors leading to student nurses’ absenteeism: student nurses being treated as a workforce in the psychiatric clinical area; shortage of staff in the psychiatric wards; inadequate supervision of student nurses in a psychiatric ward by professional nurses; and a lack of consideration toward student nurses’ requests for days off.

Regarding the students being treated as a workforce in the psychiatric wards, the study’s findings indicate that half of the respondents (50.3%) indicated that they were treated as a workforce in most of the psychiatric clinical areas. Similarly, a study conducted at the Capricorn District Nursing School found that most pupil nurses indicated that when treated as a workforce, they may have been absent from the clinical area [12]. On the other hand, ref. [19] revealed that students were not regarded as students but as additional staff not involved in patient care; thus, the senior staff are ill-treated.

A shortage of staff in the wards was identified as a critical psychiatric clinical factor leading to student nurses’ absenteeism, where 66 (8%) of the respondents indicated that the cause of absenteeism was due to a high staff shortage in the psychiatric ward. Singh [19] supports the study’s findings by revealing that student nurses were absent from the clinical area due to a shortage of staff, work overload, being treated as a workforce, and solving their own family problems.

Regarding the inadequate supervision of student nurses in the ward by professional nurses, most of the respondents, 60 (2%), indicated that supervision for students by professional nurses in the clinical areas was rarely provided. This is supported by [15], whereby the authors indicated that most students agreed that problems experienced in the clinical area, such as staff shortages, being treated as a workforce, non-supervision by qualified staff members, and large patient numbers, were reasons as to why they were absent. A lack of consideration toward student nurses’ requests for days off was also identified as leading to student nurses’ absenteeism; 56 (9%) respondents indicated that their requests for days off were not considered. This is supported by a study conducted in the United Kingdom by the authors of [20], which reported that absenteeism is associated with the nature of work, poor working conditions, the absence of regular leave arrangements, accidents, poor control, irregular transport facilities, lack of interest, indebtedness, and alcoholism and gambling habits, as well as the low level of wages.

This study was different because it was conducted on psychiatric student nurses and not on other students. Several studies have been conducted in other countries but not in South Africa, Limpopo Province, for the Limpopo College of Nursing. Most studies were conducted in theory but not in psychiatric clinical placement. In sharing the study results, mental health institutions might obtain scientific information on the factors leading to absenteeism amongst student nurses during clinical placement. They may be able to prevent absenteeism by dealing with these factors. A further study should be conducted using a qualitative approach so that a follow-up is conducted to ascertain students’ feelings on student nurse absenteeism in psychiatric clinical placement.

### Limitations

This study was restricted to three of the five districts of Limpopo Province and to only three of the five campuses of the Limpopo College of Nursing. Therefore, student nurses of the whole province were not represented well. There was a lack of relevant statistical analysis. The lack of relevant statistical analysis on the characteristics of the sample was one of the limitations to providing more rigorous results.

## 5. Conclusions

The study’s findings revealed that different factors cause student nurses’ absenteeism during psychiatric clinical placement. Most of the respondents indicated that they absent themselves from the clinical areas due to being treated as a workforce in the ward; a shortage of staff in the wards; inadequate supervision of student nurses in the ward by professional nurses; and a lack of consideration toward student nurses’ requests for days off.

Regular clinical meetings should be held between hospital managers, staff members, lecturers, and students, so clinical areas become aware of student nurses’ problems during PNS clinical placement. For this reason, managers of clinical areas will also become aware of the unfavorable working conditions and poor relations between students and professional nurses, which could lead to developing strategies to improve conditions for the students.

## Figures and Tables

**Figure 1 ijerph-20-03545-f001:**
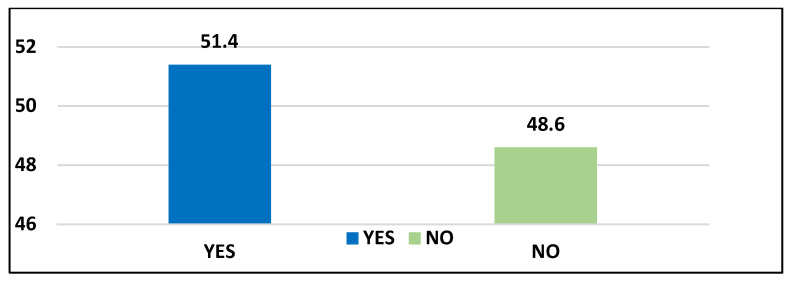
Respondents’ responses to students being treated as a clinical area workforce (N = 204).

**Figure 2 ijerph-20-03545-f002:**
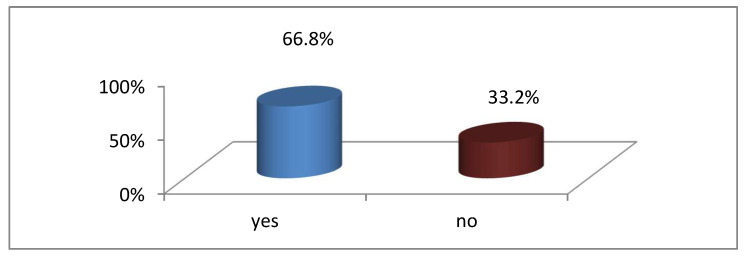
Respondents’ responses to the shortage of staff in the wards (N = 193).

**Figure 3 ijerph-20-03545-f003:**
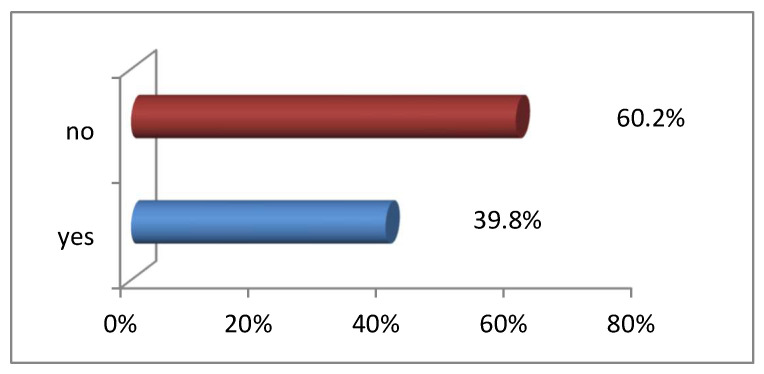
Respondents’ responses to whether there is adequate supervision for students by professional nurses (N = 196).

**Figure 4 ijerph-20-03545-f004:**
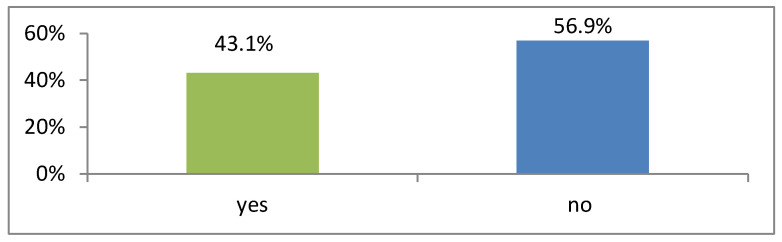
Respondents’ responses to whether their requests for days off are considered (N = 197).

## Data Availability

The anonymized data are available from the corresponding author upon request.

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
