# Peer review of "Psychiatric Clinical Placement: Agony for College Nursing Students in South Africa"

_ijerph, 2023, doi:10.3390/ijerph20043545_

Round 1
Reviewer 1 Report (Previous Reviewer 1)
The manuscript is appropriate for publication
Author Response
Thank you so much for reviewing our manuscript and thank you for the comments
Reviewer 2 Report (Previous Reviewer 2)
WHAT ARE THE PSYCHOMETRIC CHARACTERISTICS OF THIS INSTRUMENT, ITS PURPOSE AND IF VALIDITY WAS OBTAINED, IS IT UNIDIMENSIONAL OR MULTIDIMENSIONAL?
ELIMINATE TABLE 2 BECAUSE IT DOES NOT PRESENT ANY TYPE OF ASSOCIATION IN ANY OF THE VARIABLES.
FIX THE VALUES OF THE AXIS THAT REPRESENT THE PERCENTAGE VALUES OF FIGURE 1.
LOOK FOR APPROPRIATE STATISTICAL TESTS SO THAT THE CONTRAST OF THE PERCENTAGE RESULTS ALSO HAS A FORMAL MEANING.
THERE ARE MANY MORE LIMITATIONS TO THE WORK THAN ARE STATED IN IT, FROM THE ANALYSIS PROPOSAL, TECHNICAL ASPECTS OF THE INSTRUMENTS, AND THE LACK OF RELEVANT STATISTICAL ANALYSIS, ON THE CHARACTERISTICS OF THE SAMPLE, ITS REPRESENTATIVENESS, ETC., IT IS POSSIBLE THAT A BETTER JOB COULD HAVE BEEN DONE.
Author Response
Kindly check the uploaded file

Reviewer 3 Report (Previous Reviewer 3)
As this is a revised version, I am sorry to say that although the authors has made many revisions, some new problems have emerged. I do not believe the manuscript is of sufficient quality for publication. The comments are provided in the below:
1. The title of the table 2 “Association between respondents’ demographic characteristics and clinical factors leading to students’ absenteeism”. It seems quite weird that I did not see the clinical factors in this table. Please check the title whether there is mistake or not.
2. Several inferential statistics were performed in this revision, resulting in corresponding p-values. Hpwever, it is unclear from your article which statistical analyses you used. The data analysis (method section) should include these instructions in detail. That is, there must be a clear correspondence between every statistic presented in your results and the data analysis.
3. My opinion is that the current analysis of the relationship between psychiatric clinical factors and absence still cannot provide a more rigorous result. Why not directly test the association between absence with other factors? Using inferential statistics (such as regression) can complete such an analysis, which will produce a more objective result than graph comparisons.
Author Response
Kindly check the submitted file.

Round 2
Reviewer 3 Report (Previous Reviewer 3)
i have no other comments, thank you for the revision.
This manuscript is a resubmission of an earlier submission. The following is a list of the peer review reports and author responses from that submission.
Round 1
Reviewer 1 Report
I still do not find scientific soundness of this article. Please provide me comments to my first revision if that is possible
Author Response
attachment

Reviewer 2 Report
The authors clearly establish the relevance of the study, from which it is demonstrated that absenteeism in nursing students within psychiatry. However, the summary mentions the correlation between various factors and student absenteeism, but no statistical association is presented in the analysis of the data.
In the proposal of the work include a greater number of works that show the absence of studies on student absenteeism in the sections of psychiatry during the training of nurses.
Figure 1 must be corrected in the percentage values of the intervals, there are errors in which they are shown. Similarly, considering the nature of the data, it is prudent to ask why authors do not perform inferential statistical analyses (which although mentioned in the data analysis section, are not carried out).
Even the data would allow the generation of logistic regression models, with the factors questioned as predictor variables; This would mean a greater contribution of the study carried out.
Author Response
attachment

Reviewer 3 Report
Thank you for giving me this opportunity to review this manuscript, “Psychiatric Clinical Placement: An agony to College Nursing Students in South Africa”. The purpose of this study was to investigate factors leading to student nurses’ absenteeism in the Limpopo College of Nursing during the Psychiatric Nursing Science clinical placement. My comments are outlined below and I sincerely hope that the authors find them helpful in any future revisions of their work.
1. In the introduction section, it is important that the theme of the submitted papers has a global perspective rather than concentrating on a specific region or country, given the fact that the journal itself is facing the global readers. With this in mind, perhaps the authors should have considered starting with the absenteeism of nursing education, rather than introducing the situation in Limpopo province first.
2. Regarding to the factors incluecning students’ absenteeism, why you selected those variables, is there any evidence in the literature to support your selection of these variables? I suggest you to justify these in your introduction. As you don't mention these variables in your manuscript until the research methods section, the reader won't understand why you selected them.
3. It is not clear how you revised the instruemtns based on your pre-tests, Have anything been corrected or has nothing been changed, and all the questions remain the same? This information should be presented more clearly.
4. You mentioned that inferential statistics and a table were presented in the results section, but I did not see any inferential statistical results or a table.
5. As you stated that “Questionnaires consisted of 2 questions on the demographic information, which are: 142 the sample characteristics and the number of days respondents were absent in PNS clinical areas”, the number of days of absent was asked in the survey. However, it is very confusing that this should be the core variable of your research, but you did not present any results for this variable. If, as you said in the introduction, the absence is extremely serious, then the descriptive statistics of this variable will echo what you mentioned earlier. Why didn't you present it?。
6. Furthermore, to analyze the relationship between factors and absenteeism, one does not use the method commonly used in general empirical research, that is, to test the relationship among variables. Why don't you test the association between the number of days of absent and other factors? You should have enough data to do such a test, so it is really the so-called inferential statistics. The way you are doing is very uncommon, and it can easily become a subjective judgment. For example, Of the 204 respondents who responded to this question, 105 (51.4%) respondents in-190 dicated that students are treated as a workforce, while 99 (48.6%) indicated that students 191 are not treated as a workforce in the clinical areas. When the number is 51.4% or 48.6%, whichever is higher, you can't draw conclusions from comparing numerical values. At this point, inferential statistics are the only way you can get scientific results.
Author Response
attachment
